# A Comparison of Country’s Cultural Dimensions and Health Outcomes

**DOI:** 10.3390/healthcare9121654

**Published:** 2021-11-29

**Authors:** Justin C. Matus

**Affiliations:** Sidhu School of Business, Wilkes University, Wilkes-Barre, PA 18766, USA; justin.matus@wilkes.edu; Tel.: +1-570-408-4714

**Keywords:** culture, population health, health spending, life expectancy, health policy, Hofstede

## Abstract

Research comparing health care systems of countries, with a particular emphasis on health care spending and health care outcomes, has found unexplained differences which are often attributed to the countries’ cultures, yet these cultural dimensions are never completely identified or measured. This study examines if culture predicts a country’s population health, measured as life expectancy and health care spending. Using the Hofstede country-level measures (six dimensions) of culture as independent variables, two regression models to predict life expectancy and per capita health care using 2016 World Bank data were developed. The original data set included 112 countries which was reduced to a final total of 60 due to missing or incomplete data. The first regression model, predicting life expectancy, indicated an adjusted R square of 0.45. The second regression model, predicting per capita health care spending, indicated an adjusted R square of 0.63. The study suggests culture is a predictor of both life expectancy and health care spending. However, by creating a composite measure for all six culture measures, we have not found a significant association between culture and life expectancy and healthcare expenditure. The study is limited by small sample size, differences in geography, climate and political systems. Future research should examine more closely the relative influence of individualism on life expectancy and assumptions about models of socialized medicine.

## 1. Introduction

The purpose of this study is to determine if culture predicts health; specifically does a country’s culture predict the health of that country’s population. While this study will use data from 60 countries, there is an inherent emphasis on the United States for the simple reason that it is the outlier in terms of health care spending vis à vis life expectancy as further described below.

In the United States, at least, the national conversation about the cost and availability of health care is now decades old, dating back at least to President Truman in the 1950s who first proposed some type of national health insurance [1]. Specifically, much of the laments of the U.S. health care system is about the relatively high cost with a questionable return on the investment—especially as it relates to outcomes as well as the breadth or lack thereof of health care insurance coverage. Politicians and policy makers alike often look with envy at other countries such as Sweden or Denmark as models for delivering high-quality and highly cost effective health care—a duality that seems to be beyond the reach of the U.S. Put more directly, the problem appears to be that the U.S. has a very inefficient health care system. Perhaps the most damning statement was in the Commonwealth Fund’s 2017 report which read, in part, “Based on a broad range of indicators, the U.S. health system is an outlier, spending far more but falling short of the performance achieved by other high-income countries” [2].

As an example, the U.S. in 2018 spent 17.1% of gross domestic product (GDP) on health care while achieving an average life expectancy of 71 while countries like Sweden and Denmark spent 9.1% and 9.2% of GDP on health care, respectively, while achieving life expectancies of 74 and 75. Life expectancy is, of course, but one measure of health. Likewise, health spending is but one input in a health model. Further discussion of appropriate definitions and measures of health and culture will be discussed below.

So why examine culture? A simple answer might be why not? This is especially so against the backdrop of countless studies that have offered countless other explanations as to why the U.S. simply does not get the same bang for the buck as so many other wealthy countries. Nonetheless, many of these studies may only be suggestive of the symptoms of perceived inefficiencies—for example, high administrative costs or lack of universal health care insurance. Yet these explanations may in fact fail to reveal the underlying root cause or causes. In fact, as will be discussed below, often the unexplained variance in many an analysis is broadly suspected to be the ephemeral concept of culture, yet without any substantive evidence to support such claims [3,4]. In the 2017 Commonwealth Fund report which ranked the U.S. health care system lowest of the 11 high-income countries studied, the authors of the study admit in the discussion of limitations of the study that patient expectations and a physician’s assessments might be influenced by culture, without ever defining culture. 

Although the answer may lie in our culture, we are appropriately cautioned as the researchers state in their paper, *Culture: The missing link in health research*, “Culture is often proposed as an explanatory variable for these differences yet, paradoxically, little work explicates the precise cultural processes involved that are valid, relevant to the communities of focus, and sustainable” [5]. Hruschka [3] goes further, “… in most cases presented above, culture was proposed to ‘do something’ that could account for observed disparities in health … In many cases, culture was simply invoked as something that mattered for a health outcome, with no specification of particular pathway beyond broad claims about measurement issues, behaviors or cultural sensitivity”. Hruschka continues, “In a few cases, generic cultural factors were proposed as a cause of observed population differences that could not be accounted for by other means” [3].

Further support for assessing both culture and society as determinants of health are discussed at length by Hall and Lamont’s book, Successful Societies How Institutions and Culture Affect Health [6]. Once again however, problems arise when more comprehensive models are suggested with amorphous variables such as society. In his review of the aforementioned book, Huijts critiques the suggestions and writes, “Although some scholars are already actively pursuing this approach, Keating’s chapter could have benefited from some more discussion of the practical limitations associated with actually executing the comprehensive type of analyses he suggests” [7].

It is possible that the temptation to reach for culture as a plausible explanation is nothing more than an attempt at inductive reasoning. If it is not all of these other things, it must be culture. This study will attempt to advance the discussion.

### Review of the Literature

In order to answer the question of culture and health, we must first examine the two core concepts: culture and health. What is culture and what is health? Hofstede’s seminal work in cross-cultural studies posits that a country’s culture can be best understood by examining four distinct dimensions of culture [8]; they are:

Power distance: social inequality, including the relationship with authority. Individualism-collectivism: the relationship between the individual and the group. Masculinity-femininity: the social implications of having been born as a boy or a girl (later editions of the book replaced the word “social” by “emotional”). Uncertainty avoidance: ways of dealing with uncertainty, relating to the control of aggression and the expression of emotions (later editions of the book refer to “the extent to which the members of a culture feel threatened by ambiguous or unknown situations”) [9].

In 1991 Hofstede added a fifth dimension: “long-term versus short-term orientation” [9]. A sixth dimension was added in 2010 “indulgence-restraint” [10].

Hofstede’s development of a numeric score for each of his six dimensions (initially four dimensions) began in the early 1970s when, in his own words, ”by accident” he was given access to a survey data base of a very large international corporation, International Business Machines (IBM). The survey data contained individual responses from approximately 100,000 employees from more than 50 countries. This was the genesis for the further data collection and refinement of his dimensions of culture model. Based on the individual survey responses, mean scores were developed for the various dimensions for each country. Scores were standardized to a 100-point scale. These scores became the foundation for on-going research for the subsequent 40-plus years [11].

Interpreting the six dimensions and a particular country’s scores is an interesting exercise in its own right. One might wish to infer that high score is better than a low score, but this is not the intention of the scale scores. The scale scores are probably most useful when using a side-by-side comparison of one or more countries along one or more of the six dimensions. A Dane might be just as proud of a low score on power/distance, as an Italian would be proud of a high score on individuality/collective. Perhaps the best way to interpret the scores is to think of them as a pair of semantic differences and the mid-point is 50. A score that is either above or below the mid-point simply helps one understand the relative strength of a country’s particular measure on a particular dimension.

Hofstede’s model has been used in numerous studies and settings for a variety of purposes. There are also more than a few cautionary tales of Hofstede’s model being misused [12,13,14]. Nevertheless, Hofstede’s model, when used appropriately has been in the literature for well over 40 years and cited numerous times. The single greatest concern about the model is when researchers attempt to ascribe country-level characteristics to individuals, making the so-called ecological fallacy error. Hofstede discusses this problem at length as have many others [11,12,13]. The simple point of the ecological fallacy error is that in a sense the Hofstede model can be a blunt instrument if not deployed properly and one should proceed with caution. Hofstede warns, “One of the weaknesses of much cross-cultural research is not recognizing the difference between analysis at the societal level and the individual level amounts to confusing anthropology and psychology. From 180 studies using my work reviewed by Kirkman, Lowe and Gibson, more than half failed to distinguish between societal culture-level and individual-level differences, which led to numerous errors of interpretation and application” [11,14]. Logically, culture is never about a single person, but rather a collection of individuals—as Hofstede’s states, “Culture is the collective programming of the mind that distinguishes the members of one group or category of people from others”. Even more precisely, Hofstede writes, “… culture and personality are linked but the link is statistical; there is a wide variety of individual personalities within each national culture and national culture scores should not be used for stereotyping individuals” [11].

Singer, Dressler, George and et.al. addressed culture in a health care context, describing culture, at least in part, as follows:

Culture is an internalized and shared schema or framework that is used by group (or subgroup) members as a refracted lens to “see” reality, and in which both the individual and the collective experience the world. This framework is created by, exists in, and adapts to the cognitive, emotional and material resources and constraints of the group’s ecologic system to ensure the survival and wellbeing of its members, and to provide individual and communal meaning for and in life [5].

In their discussion of the current state of culture and health research, they further posit that, “Culture is often operationalized with superficial, simplistic and crude measures, such as dichotomous nominal variables based ostensibly on race or singular, stereotypical beliefs … When such variables are entered into statistical analyses as proxies for culture, the findings are inconclusive or, at best, contribute negligible explanatory weight to the variance of health outcomes”. They conclude that to operationalize culture in a health research setting that the “approach could either be more of a discovery or formative research mode, or from a well-defined theoretical orientation identifying specific cultural constructs of interest” [5]. The Hofstede model offers such a theoretical framework.

The second key concept of this study is health, specifically the health of a country. Just as the concept of a country’s culture might at first seem intuitive yet ultimately prove elusive so too is the concept of a country’s health. While The World Health Organization (WHO) definition, “Health is a state of complete physical, mental and social well-being and not merely the absence of disease or infirmity” has not changed and may work well if thinking about an individual’s health status, it may not apply so easily when thinking about the health of a nation [15]. In fact, the WHO 2018 report lists 35 different health indicators in their worldwide health assessment [16]. Accordingly, this study will look at several characteristics that speak to the health of a community, the communities in this case being the countries in question.

There have been countless studies which have sought to compare various country’s models of health care systems, their financing mechanisms and most importantly their health outcomes. Stated simply, for many policy makers a nation’s health is a simple function of the net gain achieved from inputs versus outputs. The more you put into a system, the more you get out. Unfortunately, this logic does not hold up very well, and is most acutely observed when one looks at the percent of GDP, spent on healthcare in the United States, in 2018 the world’s highest of 17.1%, while its average life expectancy of 78.5, a common measure of population health, ranks only 53rd highest in the world, well behind other countries with much lower rates of spending and much longer life expectancies [16].

McDowell, Spasoff and Kristjansson in 2004 examined the challenge of defining and measuring population health. They suggest there are two models, the first labeled as the descriptive model, wherein population health is measured and defined by measures such as life expectancy, which may then be stratified by other markers such as race or socio-economic status [17]. They state that, “this approach to defining population health is limited and does not adequately capture its scope as an academic field of study”. To ameliorate this shortcoming, they proffer a second model, the analytic model. The analytic model, “refers to a conceptual and analytic approach to explaining why some people are healthy and others are not, at its broadest, it seeks to analyze not only how this occurs, but why. This conception demands a much broader measurement protocol that includes not only outcome variables in terms of morbidity and mortality indexes, but also direct measures of health processes within our population”.

Anderson and Frogner evaluated health spending in the countries of the Organization for Economic Cooperation and Development (OECD). The OECD gathers self-report data from its member countries affording researchers a rich source of data, although the quality and/or reliability of the data has been sometimes questioned [18,19]. In Anderson and Frogner’s study, they were particularly interested in “value per dollar”. Their methodology defined health care quality as life expectancy and value was calculated via a “bivariate regression of life expectancy and GDP per capita (measured in PPP) [18]. The OECD explains, “PPPs are the rates of currency conversion that eliminate the differences in price levels between countries. Per capita volume indices based on PPP-converted data reflect only differences in the volume of goods and services produced. Comparative price levels are defined as the ratios of PPPs to exchange rates. They provide measures of the differences in price levels between countries. The PPPs are given in national currency units per US dollar” [20]. Anderson and Frogner while admitting that, “The relationship between life expectancy and health spending is a crude analysis of value per health care dollar”, readily recognize that “many other factors influence life expectancy and life expectancy is only one measure of health status. They wrote, “… we have returned to the conclusion that much of the spending differences are attributable to the higher per capita income of the United States and the fact that Americans pay much higher process for medical care services …”. Nevertheless, the researchers also examined 16 health care quality indicators across the 29 OECD countries in their study at that time and concluded, “… the results suggest the United States is not receiving good value for a country that spends more than any other OECD country. Overall, the United States is about as likely to be in the top half as in the bottom half of the countries submitting data”. For example, for cervical cancer screening, the U.S. had the highest rank at number 1, for the indicator asthma mortality rate, the U.S. ranked number 23 out 25 countries reporting [21].

A more recent study by Bradley, Sipsma and Taylor used a mixed methods approach using “a 10-year analysis of national-level health and social service spending and health outcome data from the OECD”. The title of their paper is in itself revealing: American Paradox—high spending on health care and poor health. In this study the researchers examined not only spending on medical services but also the ratio spent on social services. They write, “The academic literature is unequivocal about the importance of what is referred to as the ‘social determinants of health’ such as housing, education, income, occupation, environmental contacts, lifestyle and other similar exposures”. Their findings indicated that the U.S. ratio of spending on social service to health care was the lowest amongst all the OECD countries. Conversely, when health and social service spending are totaled, the U.S. ranks in the middle. They found that, “using health and social expenditure data from 2009 and mixed effects regression models, we find countries with higher ratios of social-to-health spending have significantly higher life expectancy and lower infant mortality after adjusting for health expenditures and GDP”. Bradley, Sipsma and Taylor conclude that, “… the social determinants of health are essential to the lives of Americans from a range of backgrounds and income levels”. One possible conclusion from this study is that while the total spending of the U.S. is not abnormal, the proportion of spending on health vis à vis social services is [22]. Heuvel and Olariou in a study of European countries examining GDP health spending and life expectancy, found that spending was not a main determinant of life expectancy, however spending a higher proportion of GDP on social protections contributed to longer life expectancy [23].

In 2019, Anderson and Poullier examined health spending and health outcomes across OECD countries. In spite of the long-running dialogue of what to measure and how, progress has seemingly been slow. Their study, like many before, including some mentioned above, used PPP measures (as inputs), rates of health insurance coverage, hospital admission rates and physician visits, to name a few, as process measures and of course, life expectancy as health outcomes. They continued to recognize the many problems in such studies. “Comparisons of outcomes using international data are fraught with problems. It is widely recognized that most standard outcome measures such as longevity or infant mortality are only crude proxies for health status …” [24]. The WHO in their 2018 report, Monitoring Health for SDG’s (Sustainable Development Goals) likewise cautioned: “Health data derived from health information systems, including health facility records, surveys, or vital statistics, may not be representative of the entire population of a country and in some cases may not even be accurate. Comparisons between populations or over time can also be complicated by different data definitions and/or measurements” [16]. 

## 2. Methods and Data

This study will utilize country-level data from the Hofstede center’s web site on 112 countries including measures for the six dimensions of culture described earlier (Hofstede 2020). This data set will be matched with the same countries from the World Bank to include GDP health care spending, per capita health care spending and life expectancy. World Bank health care spending is reported in adjusted dollars allowing for country to country comparisons. A final data set will be collated by matching cases with complete data for all variables of interest. Statistical analysis and modeling will be done using SPSS software (IBM, Armonk, NY, USA). The following research questions will be investigated:

Q1. Do the dimensions of culture predict life expectancy?

Q2. Do the dimensions of culture predict health care spending?

Data was collected on the 112 countries from the Hofstede web site in an Xcel file [25]. Data sets for percent of GDP health care spending, per capita health care spending and life expectancy were downloaded from the OECD web site [20]. The most recent OECD data available were for the year 2016.

An example of the six dimensions of culture data is shown in Figure 1.

Selected countries in Figure 1 were chosen to simply highlight some of the more distinctive contrasts for illustrative purpose only and no particular interpretation is implied. The first item, PDI, is the power-distance scale. Hofstede defines this as, “the extent to which the less powerful members of institutions and organizations within a country expect and accept that power is distributed unequally. The second item. IDV, is the individualism scale, defined as, “the degree of interdependence a society maintains among its members”. The third item, MA, masculinity, is defined as “the fundamental issue here is what motivates people, wanting to be the best (masculine) or liking what you do (feminine)”. The fourth item, UAI, uncertainty avoidance, is defined as, “the way that a society deals with the fact that the future can *never be known*”. The fifth item, LTO, long-term orientation, is defined as, “how every society has to maintain some links with its own past while dealing with the challenges of the present and future”. The sixth item, INDR, Indulgence-Restraint, is defined as “the extent to which people try to control their desires and impulses” [26].

The U.S. scores are relatively higher on Individualism and Indulgence and relatively lower on power-distance and long-term orientation. U.S. scores for Masculinity and uncertainty avoidance are somewhat moderate. Hofstede interprets these U.S. scores, in part, by stating, “… The society is loosely-knit in which the expectation is that people look after themselves and their immediate families only and should not rely (too much) on authorities for support” [26].

Figure 2 provides per capita health spending by select OECD country. Figure 3 provides health spending as a percent of GDP by select OECD country [20]. Selected countries in Figure 2 and Figure 3 were chosen to simply highlight some of the more distinctive contrasts for illustrative purposes only and no particular interpretation is implied.

Figure 4 presents 2016 average life expectancy by selected countries. Selected countries in Figure 4 were chosen to simply highlight some of the more distinctive contrasts for illustrative purpose only and no particular interpretation is implied.

A final data set was developed after by merging the Hofstede data and the World Bank data. A total of 60 countries comprised the final data set. Several countries were dropped from the analysis because of incomplete data for several of the cultural dimensions. Two regression models were developed, one each for the two dependent variables, life expectancy and per capita health care spending. GDP health care spending was not used in the analysis since it is a relative measure whereas per capita health care spending is an absolute measure.

The six dimensions of culture were entered as independent variables. The six dimensions of culture are: power-distance (PD); individualism-collectivism (IDV); masculinity-femininity (MF); uncertainty avoidance (UA); long-term versus short-term orientation (LTO); indulgence-restraint (IND). The six dimensions are measured on scale from zero to 100, the higher the score the greater the dimension’s influence.

There are several concerns in this study with such a particularly small sample size for a regression model. Field suggests that uncovering a larger effect may not suffer as greatly as when the expected effect is relatively smaller. The problem is further compounded by the number of predictor variables, in our case there are six (dimensions of culture). Nevertheless, a sample size of sixty while at the very edge of a lower limit with a model having six predictor variables, there is modest support for proceeding [27]. Indeed, an N as low as >25 has recently been suggested by Jenkins and Quintana-Ascencio as sufficient for regression [28]. This issue will be discussed further in the limitations section below.

## 3. Results

The first regression model examined life expectancy. Descriptive statistics are presented in Table 1 below.

Table 2 presents the regression results. The model’s adjusted R square equaled 0.45, with three of the six independent variables entering the equation at a significance level *p* < 0.01. The variables and their respective Betas were: IDV = 0.37; LTO = 0.41; IND = 0.32. These results indicate the impact of culture on life expectancy, specifically the influence of individualism, long-term orientation and indulgence. Individualism versus the collective, as Hofstede has described, is the degree to which one puts their own interests above others [11]. In our model, a higher level of individualism indicates a longer life expectancy. Long-term orientation versus short term orientation suggests that higher scores are indicative of societies which are more future-oriented whereas lower scores indicate a desire to hold on to customs and traditions. The inference in the latter case may be that these societies are slower to change and adapt to their environments which in turn suggests a shorter life expectancy. Finally, the indulgence versus restraint dimension suggest that a higher score is indicative of a society where people are more likely to reward themselves whereas cultures with restraint (a lower score on the scale) are apt to conform to societal norms. The variables which did not enter the model were power distance, masculinity/femininity and uncertainty avoidance.

The second regression model examined per capita health care spending. Descriptive statistics are presented in Table 3 below.

Table 4 presents the regression results. The adjusted R square equaled 0.63, with four of the six independent variables entering the equation at a significance *p* < 0.01. The variables and their respective Betas were: PDI = −0.31; IDV = 0.39; LTO = 0.32; IND = 0.34. These results indicate the strong impact culture has on health spending. In particular, the degree of individualism, long-term orientation and indulgence suggest an increase in health spending, whereas a higher measure of power distance suggests lower spending. Like the first model, individuality, long-term orientation and indulgence were predictive of health spending. The fourth variable to enter the equation is power distance with a negative beta of −0.314. The addition of this fourth variable has also increased the adjusted R square to 0.63 compared to the first model with an adjusted R square of 0.45. Power distance relates to the relative acceptance of power, i.e., government power, being concentrated at the top or more disbursed or more evenly distributed among the population. In this case, the higher the power distance score, the less health care spending.

VIF values and tolerance statistics for both regression models were also calculated. There were no VIF values greater than 10. Likewise tolerance statistics were all above 0.2, indicative of no multicollinearity. Further tests for assumptions were performed by plotting standardized residuals against standardized predicted values. Visual inspection of the plots suggested that assumptions of linearity, normality and homoscedasticity have been met. Partial regression plots of the six independent variables were also performed for both models. Once again, visual inspection of the plots suggested that assumptions of linearity and homoscedasticity have been met [27].

## 4. Discussion

The first research question, do the dimensions of culture predict life expectancy, was addressed in the first regression model described above. There is moderate evidence that three of the six dimensions of culture influence life expectancy, specifically individualism, long term orientation and indulgence. At this juncture, it is important to remember that we do not make the ecological fallacy error of ascribing the country level characteristics to any one individual’s behavior as described by McSweeney [14]. Put simply, we should not conclude that if a person were to be more indulgent they might live longer—although that would be a comforting rationalization to buy a new car or take a vacation. On the other hand, the relative influence of individualism on life expectancy in this model may give rise to assumptions about models of socialized medicine. One may be tempted to argue from this particular result that an “every person for themselves” outlook is a rational one and leads to a longer life. Conversely, an argument may be made that collectivism leads to a shorter life span, neither of which we are suggesting. What we do suggest is that culture, as defined and measured by Hofstede, does have an influence on a country’s health status as measured here in terms of life expectancy.

The second research question, do the dimensions of culture predict per capita health care spending, was addressed in the second regression model above. There is moderate evidence that four of the six dimensions influence per capita health care spending. Specifically individualism, long term orientation and indulgence suggest an increase in health spending, whereas a higher measure of power distance suggests lower spending. Similar to the first regression model, individualism, long-term orientation and indulgence contribute significantly to the overall model in predicting per capita health care spending. Additionally, the fourth dimension, power distance, also enters the model and has a negative relationship to health care spending. In a culture where people are more accepting of power being unevenly distributed, there is a concomitant likelihood of lower per capita health care spending. This particular finding is interesting in that we wonder whether in cultures which expect less of the holders of power, they do in fact receive less. It begs the question as to whether it is a self-fulfilling prophecy. An argument might be made that in a country like the United States, more money is spent on health care per capita because leaders are held more accountable to spend more, but not necessarily spend wisely.

### 4.1. Post Hoc Sensitivity Analysis

A post hoc sensitivity analysis was performed. The six dimensional scores were transformed into a single new variable labeled “Total Culture”. Each of the six dimension scores were summed to calculate the Total Culture score. The range and mean of the scaled score were calculated (range = 235–426, mean = 318). The Total Culture score was then recoded into a second new variable labeled Total Culture Scaled (TCS) along five quintiles from very low to very high, shown in Figure 5. Once again, two regression models were performed. The first model, has TCS as the independent variable and life expectancy as the dependent variable, with results displayed in Table 5. The second model again has TCS as the independent variable and health care spending per capita as the dependent variable, with results displayed in Table 6.

The first model, life expectancy, resulted in an adjusted R square of 0.035 and the TCS with a significance level of 0.08.

The second model, health spending, resulted in an adjusted R square of 0.01 and the TCS with a significance level of 0.21.

The lack of a significance levels below 0.05 for TCS in both models and fairly low adjusted R squares may further suggest weakness in the original models. However, the collapsing of the six dimensional scores may be masking one or more of the dimensions’ predictive power inasmuch as a given country’s high score along one dimension may effectively cancel out another lower score on a second dimension. It also possible that dimensionality may mask one more of the more predictive independent variables [29,30]. However, by creating a composite measure for all six culture measures, we have not found a significant association between culture and life expectancy and healthcare expenditure. Future studies should seek larger sample sizes as well as controls for other variables such as population density or gross domestic product.

The above findings should be useful to researchers concerned with the influence of culture on a population’s health, but also to what extent, if any, various dimensions have on other aspects of society. For example, several studies on military spending have looked at the effect on GDP and debt [31,32]. Questions might be: is there a parallel effect of culture on military spending? Does power-distance predict military spending? Does restraint-indulgence predict spending on education or educational attainment?

While there no doubt remain many unanswered questions about the influence of culture on health at the individual level as well as the country level, this foundational research sets up further investigation with perhaps a focus on the various singular dimensions of culture. There are also competing models that attempt to measure culture, such as GLOBE, the Schwartz cultural values framework, and the World Values Survey, and these too may further the discovery of more powerful predictors [33,34,35].

### 4.2. Limitations

The sample size of 60 countries is very small, given the number of independent variables (six). Whether it is appropriate to consider only Hofstede’s model in its totality versus breaking it into its component parts is a matter for discussion. This research did not take directly into account nor control for a country’s wealth, population size, climate or form of government; all of which could be important contributors to understanding both culture and health. An open question also remains as to the fluidity of a country’s culture. Hofstede states, “National Culture cannot be changed, but you should understand and respect it” [26].

## 5. Conclusions

This study demonstrates that Hofstede’s model of a country’s culture predicts the level of per capita health care spending and life expectancy. More specifically, the particular dimensions of culture such as individualism, indulgence, power-distance and long term orientation all contribute significantly and might give rise to a deeper understanding of a country’s health status and health care spending. The implications for policy makers in the United States in particular may be to make an appeal based on the economic benefits of a collective model of health care rather than making an emotional appeal or an argument based one’s social responsibility.

## Figures and Tables

**Figure 1 healthcare-09-01654-f001:**
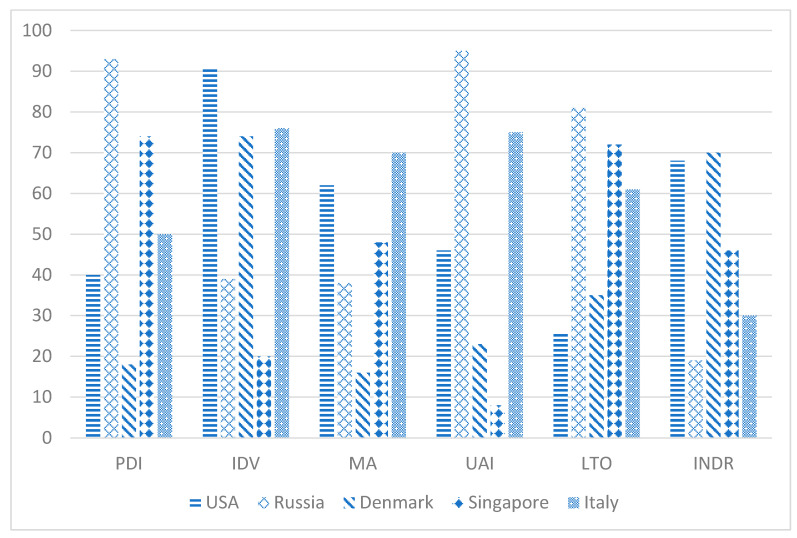
Examples of six dimensions of culture.

**Figure 2 healthcare-09-01654-f002:**
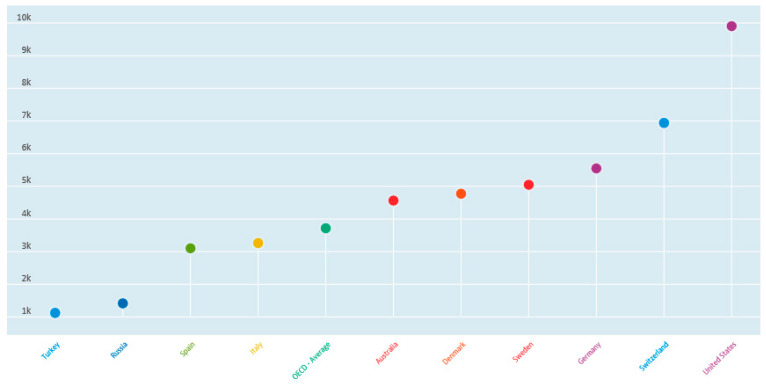
Per capita healthcare spending by country.

**Figure 3 healthcare-09-01654-f003:**
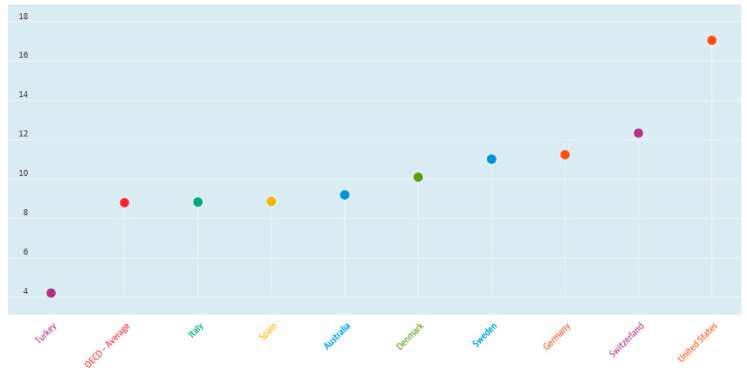
Per cent of gross domestic product (GDP) healthcare spending by country.

**Figure 4 healthcare-09-01654-f004:**
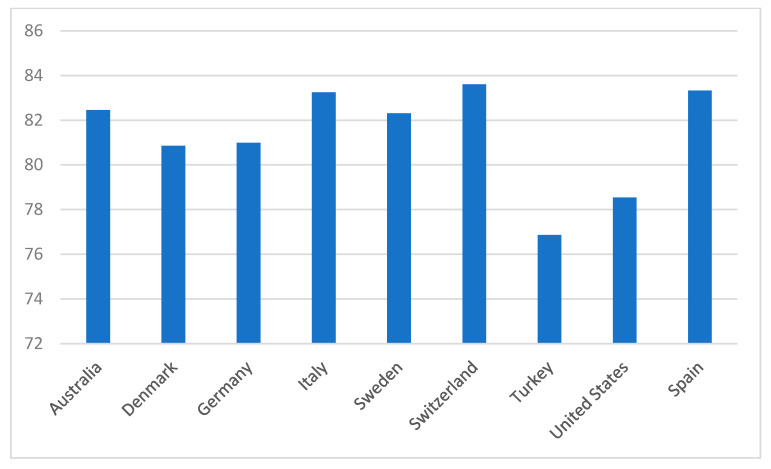
Life expectancy in 2016.

**Figure 5 healthcare-09-01654-f005:**
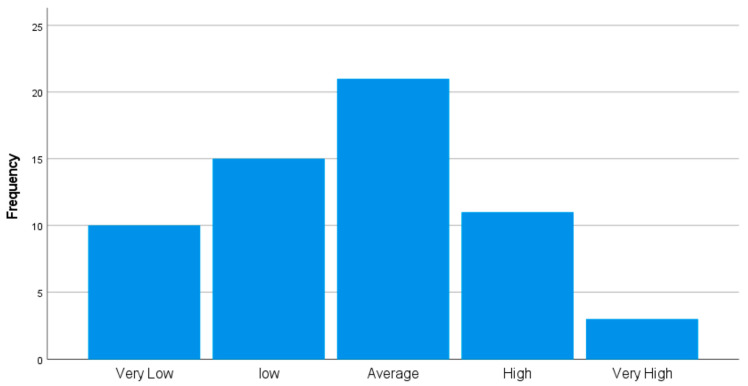
Total Culture Scaled.

**Table 1 healthcare-09-01654-t001:** Life Expectancy Model 1: Descriptive Statistics.

Variables	Mean	St. Deviation	*N*
Life Expectancy	77.73	4.44	60
pdi	58.4833	20.94	60
idv	46.0167	23.91	60
ma	49.1000	20.57	60
uai	67.2667	22.86	60
lto	49.5928	23.21	60
ind	48.1013	22.37	60

pdi = power-distance; idv = individual-collective; ma = masculine-feminine; uai = uncertainty avoidance; lto = long term orientation; ind = indulgence-restraint.

**Table 2 healthcare-09-01654-t002:** Life Expectancy Model 1: Regression results.

Model		Unstandardized Coefficients			Standardized Coefficients	t	Sig.
		B		Std. Error	Beta		
(Constant) Life Exp.		69.440		3.585		19.367	0.000
pdi		−0.045		0.029	−0.213	−1.542	0.129
idv		0.070		0.024	0.374	2.845	0.006
ma		−0.016		0.022	−0.076	−0.745	0.459
uai		0.023		0.020	0.119	1.189	0.240
lto		0.079		0.022	0.413	3.567	0.001
ind		0.064		0.024	0.32	2.627	0.011
**Adj R Square**	**Std Error of Est.**	**R Square Change**	**F Change**	**Df1**	**Df2**	**Sig F Change**	**Durbin-Watson**
0.45	3.31	0.50	8.87	6	53	0.000	1.84

pdi = power-distance; idv = individual-collective; ma = masculine-feminine; uai = uncertainty avoidance; lto = long term orientation; ind = indulgence-restraint. Df1 = numerator degrees of freedom; Df2 = denominator degrees of freedom.

**Table 3 healthcare-09-01654-t003:** Per Capita Health Care Spending Model 2: Descriptive statistics.

Variables	Mean	St. Deviation	*N*
Per Cap Health Care Spend	2606.48	2185.00	60
pdi	58.4833	20.94	60
idv	46.0167	23.91	60
ma	49.1000	20.57	60
uai	67.2667	22.86	60
lto	49.5928	23.21	60
ind	48.1013	22.37	60

pdi = power-distance; idv = individual-collective; ma = masculine-feminine; uai = uncertainty avoidance; lto = long term orientation; ind = indulgence-restraint.

**Table 4 healthcare-09-01654-t004:** Per Capita Health Care Spending Model 2: Regression results.

Model		Unstandardized Coefficients			Standardized Coefficients	t	Sig.
		B		Std. Error	Beta		
(Constant)		37.217		1446.308		0.026	0.980
pdi		−32.794		11.812	−0.314	−2.776	0.008
idv		36.027		9.863	0.394	3.653	0.001
ma		1.590		8.898	0.015	0.179	0.859
uai		−5.063		7.872	−0.053	−0.643	0.523
lto		30.443		8.941	0.323	3.405	0.001
ind		32.889		9.762	0.337	3.369	0.001
**Adj R Square**	**Std Error of Est.**	**R Square Change**	**F Change**	**Df1**	**Df2**	**Sig F Change**	**Durbin-Watson**
0.63	1336.09	0.664	17.47	6	53	0.000	1.84

pdi = power-distance; idv = individual-collective; ma = masculine-feminine; uai = uncertainty avoidance; lto = long term orientation; ind = indulgence-restraint. Df1 = numerator degrees of freedom; Df2 = denominator degrees of freedom.

**Table 5 healthcare-09-01654-t005:** Life expectancy.

Adj R Square	Std Error of Est.	R square Change	F Change	Df1	Df2	Sig F Change	Durbin-Watson
0.035	4.3655	0.052	3.167	1	58	0.080	1.708

Df1 = numerator degrees of freedom; Df2 = denominator degrees of freedom.

**Table 6 healthcare-09-01654-t006:** Health spending.

Adj R Square	Std Error of Est.	R square Change	F Change	Df1	Df2	Sig F Change	Durbin-Watson
0.009	2180.77	0.026	1.536	1	58	0.220	1.807

Df1 = numerator degrees of freedom; Df2 = denominator degrees of freedom.

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
