# Peer review of "A Comparison of Country’s Cultural Dimensions and Health Outcomes"

_healthcare, 2021, doi:10.3390/healthcare9121654_

Round 1

Reviewer 1 Report

This paper sets out to address the question as to whether “culture predicts the country’s population health, measured as life expectancy and health care spending.”. The author concludes that “culture predicts the country’s population health, measured as life expectancy and health care spending’. 

While the author has raised some interesting issues, two key questions must be addressed before publication is considered although to his credit, he has explored both in the introduction. 

The first, is the term ‘culture’. The author has provided a useful discussion on the meaning of culture and the difficulties in translating what could be found at a country level and what exists within that country. This creates a problem when trying to present a measure of culture. Furthermore, the culture of a society alters over time not least within an individual’s lifetime. For example, the levels of migration, political developments and generational differences will all affect it. Furthermore, there are elements within a society which are not governed by culture. These include how leaders are chosen, the arrangements for governance,  and the legal framework. One example is foreign policy, which is determined by the state nation not culture. An example of this and change is in the United Kingdom and Brexit. 

The second issue centres on the definition of health. The current title of the paper talks about health outcomes. What data the author has used is in the vast majority associated with treatment provision. This is one aspect of factors that influence health; the determinants of health and into a large extent, the outcomes of health treatments, lie outside of the health care system. As such direct comparisons are difficult and great cre needs to be taken. This issue needs expanding in the current paper.

There is an interesting and useful paper in the making but currently further work 
in the discussion section is required. Two articles which the author may find use are:

•    Successful societies: how institutions and culture affect health (2009)
Eds: Hall PA and Lamont M Cambridge University Press: New York

•    Trujillo MD and Plough A. (2016) Building a culture of health: A new framework and measures for health and health care in America Social Science & Medicine 165:206-213

Author Response

Thank you for your comments and suggestions. 

I understand your comments about culture, both in terms of measuring the construct and the question of how enduring a country’s culture is over time.  Hofstede’s work is the seminal attempt at doing just that, measuring the country’s “culture” along the dimensions he has defined.  Hofstede also holds that a country’s culture while not static, evolves over generations, and not necessarily in a discrete way, i.e., an influx of refugees would necessarily equate with a shift in that country’s culture.  Hofstede states, “National Culture cannot be changed, but you should understand and respect it”.

I do not disagree that society has other influences, but the point of the study is to only look at the relationship of culture, as defined in the study, to the single health outcome of life expectancy and the single health input, per capita healthcare spending.  As I discuss in the introduction the missing link in the literature is this examination of culture’s influence on a country’s health.   I am not at all suggesting that culture is sole determinant of what constitutes a country’s society or its health status.  I believe I say as much at the end of the paper. 

The suggestion of different and perhaps more comprehensive measures of health is problematic since finding a common measure using the same metrics across sixty or more countries creates a new set of problems in terms of internal validity, so I would not choose to go down this path.  I agree that direct comparisons of one country’s life expectancy with another’s is fraught with some issues, whether it’s the wealth of a country, the political system, geography, accessibility to healthcare, etc., etc.  While life expectancy may not be the most elegant measure, it is the most succinct and ultimate measure of health, is it not?  “Comparisons between populations or over time can also be complicated by different data definitions and/or measurements” (WHO, p. 1).  The source you suggest offers support as well, “On the other hand, health is a relatively uncontroversial measure of well being – longer life expectancies and lower rates of mortality can reasonably be associated with the success of a society’. (Hall & Lambert, 2009, p.2).  I also did not find anything useful or germane in the other citation you mentioned, Trujillo MD and Plough A. (2016) Building a culture of health: A new framework and measures for health and health care in America Social Science & Medicine 165:206-213.

In summary you bring up some interesting points, but for the most part I will have to disagree with your take – if I were to go anywhere near some of the suggestions I would be going down a rabbit hole doing a completely different study.  I will point out that I did in fact cite the work of Hall & Lamont, but I found that the essays were again much more about society than culture and I think at times the terms as applied are conflated.  Thank you again for taking the time to review and offer comments.

References

World health statistics 2018: monitoring health for the SDGs, sustainable development goals.                   Geneva: World Health Organization; 2018.   Licence: CC BY-NC-SA 3.0 IGO.

Successful societies: how institutions and culture affect health (2009)
Eds: Hall PA and Lamont M Cambridge University Press: New York

Reviewer 2 Report

This is an interesting study and the author has examined the effect of culture on life expectancy and per capita health spending. the manuscript is generally well written and could be very enlightening on linking the health and culture.  However, in my opinion the manuscript has some shortcomings as below and revising them can improve it.

1-  although the manuscript utilizes its data from numerous countries but the introduction section mainly elaborates the situation of the U.S., and at first look it seems that the study is trying to only examine the U.S. but the data for 60 countries have been analyzed. the introduction needs to be matched with the whole sample.

2-as the author mentions the sample size is small. why the author does not use the Panel data design (cross sectional and longitudinal) to increase the sample size.

3-one of the assumptions that usually is being assessed for linear regression is normality. it would be better if the author check this assumption.

4-in the discussion section, the relevancy of the results of this study with other available studies in this regard needs to be discussed.

Author Response

1-  although the manuscript utilizes its data from numerous countries but the introduction section mainly elaborates the situation of the U.S., and at first look it seems that the study is trying to only examine the U.S. but the data for 60 countries have been analyzed. the introduction needs to be matched with the whole sample. 

Responses: I changed the introduction.

2-as the author mentions the sample size is small. why the author does not use the Panel data design (cross sectional and longitudinal) to increase the sample size. 

Responses: I would be trading one set of problems for another – also, life expectancy does not change very much year over year. Likewise, because I am using per capita spending (instead of percent of GDP) those numbers while less stable than life expectancy, I would not expect huge changes over say a 3 year period.  Ultimately I will do a deeper dive, perhaps using only one or two of the variables identified in this study for which there is more country level data (Hofstede’s institute has data on more than 60 countries, but for many of those countries they only have 1 or 2 of the dimensions measured).

3-one of the assumptions that usually is being assessed for linear regression is normality. it would be better if the author check this assumption.

Responses: This had been analyzed but was omitted in my data discussion. It has been added.

4-in the discussion section, the relevancy of the results of this study with other available studies in this regard needs to be discussed.

Responses: I added some further context to the discussion, but I prefer not to re hash much of what is in the lit review.  The data speaks for itself.  I don’t think there is any comparable studies that I would point to and say, well my results confirm this previous research. 

I thank you for your review and comments.  Thank you again for taking the time to review.